# The Israeli Experience with the “Green Pass” Policy Highlights Issues to Be Considered by Policymakers in Other Countries

**DOI:** 10.3390/ijerph182111212

**Published:** 2021-10-26

**Authors:** Ruth Waitzberg, Noa Triki, Sharon Alroy-Preis, Tomer Lotan, Liat Shiran, Nachman Ash

**Affiliations:** 1Department of Health Care Management, Faculty of Economics & Management, Technische Universität, 10623 Berlin, Germany; 2Department of Health Policy and Management, School of Public Health, Faculty of Health Sciences, Ben-Gurion University of the Negev, Beer-Sheva 8410501, Israel; 3The Smokler Center for Health Policy Research, Myers-JDC-Brookdale Institute, Jerusalem 91037, Israel; 4Ministry of Health, Jerusalem 9101002, Israel; noa.triki@moh.gov.il (N.T.); nachmana@moh.gov.il (N.A.); 5Public Health Services, Ministry of Health, Jerusalem 9101002, Israel; sharon.alroy@moh.gov.il (S.A.-P.); liat.shiran@moh.gov.il (L.S.); 6Ministry of Public Security, Jerusalem 9103401, Israel; tomer.lotan@gmail.com; 7Department of Health Systems Management, School of Health Sciences, Ariel University, Ariel 40700, Israel

**Keywords:** vaccine certificate, COVID-19, public health policy

## Abstract

In the first half of 2021, Israel had been ahead of other countries concerning the speed of its rollout and coverage of COVID-19 vaccinations. During that time, Israel had implemented a vaccine certificate policy, the “Green Pass Policy” (GPP), to reduce virus spread and to allow the safe relaxation of COVID-19 restrictions in a time of great uncertainty. Based on an analysis of GPP regulations and public statements compiled from the Israeli Ministry of Health website, we describe the design and implementation of the GPP. We also look back and discuss lessons learned for countries that are considering a GPP policy, given the current upsurge of the Delta variant as of summer 2021. To reduce equity concerns when introducing a GPP, all population groups should be eligible for the vaccine (contingent on approval from the manufacturer) and have access to it. Alternatively, health authorities can grant temporary certificates based on a negative test. We also highlight the fact that in practice, there will be gaps between the GPP regulations and implementation. While some places might require a GPP without legal need, others will not implement it despite a legal obligation. The GPP regulations should have standardised epidemiological criteria, be implemented gradually, remain flexible, and change according to the epidemiological risks.

## 1. Introduction

The COVID-19 pandemic has forced many countries to impose lockdowns and restrictions on their residents to control the surge of this disease [1]. After more than one year of restricted life, bringing high economic and social burdens, COVID-19 vaccines have allowed countries to relax some restrictions and return to a more or less normal life. While most people are eager to do so, the path is uncertain and should be planned with caution. The use of vaccination certificates is not new. Green pass policies have been already used in some countries to restrict the entrance of children without infant immunisation to kindergartens [2,3]. Others require vaccine passports (e.g., for yellow fever) for foreign visitors [4]. While contentious, many countries have been long considering the implementation of vaccine certificates for different purposes [5]. Among others, South Korea, Austria, Denmark, the UK, New York (voluntarily), Germany, France, and Estonia are already implementing GPPs to allow the relaxation of some COVID-19 restrictions [6,7,8,9,10]. On 1 July 2021, the European Commission (EU) launched a vaccine passport to facilitate safe free movement within the EU during the COVID-19 pandemic for those who are vaccinated, recovered, or negatively tested [11]. 

By early May 2021, Israel had vaccinated about 90% of its residents aged 16+ and was ahead of other countries in terms of vaccination coverage and a return to normal life [12]. Israel was also a pioneer in establishing a vaccine certificate policy, also known as the “Green Pass Policy” (GPP), with the main objective, at that point, of allowing safe relaxation of COVID-19 restrictions. The rationale was that the GPP significantly reduced the risk of disease and its transmission, as only those who were vaccinated, recovered, or had a negative test were allowed to return to normal life. It was implemented for about four months, from 21 February to 1 June 2021. In mid-May, the government decided to lift the GPP in view of the high vaccination coverage along with consistent low and declining morbidity and mortality rates. However, on 21 July, Israel gradually reintroduced the GPP as a response to the upsurge from the Delta variant. 

The aims of this paper are to describe the design and implementation of the GPP between February and June 2021 and analyse how it overcame some of the ethical and practical drawbacks mentioned in the literature. We discuss and highlight previously unexplored issues that policymakers should consider when planning a GPP policy as a dynamic public health tool. 

## 2. Materials and Methods

This study takes a qualitative approach. The materials used are the GPP regulations and public statements compiled from the MoH’s press release website (https://www.gov.il/en/departments/news/?OfficeId=104cb0f4-d65a-4692-b590-94af928c19c0&skip=0&limit=10) (accessed on 18 October 2021). Data were collected between November 2020 and September 2021 and serves as the primary source for this article. Some content has been supplemented with information from the Health System Response Monitor (HSRM, https://www.covid19healthsystem.org/) (accessed on 18 October 2021), an online platform established in March 2020 in response to the COVID-19 outbreak to collect and organise up-to-date information on how countries in the WHO European region have been responding to the crisis. The methods entailed first summarising the regulations and the relevant documents related to the GPP and then analysing the facilitators of the implementation and enforcement of the GPP in Israel while relating to ethical and equity issues. 

## 3. Results 

### The Israeli Green Pass Policy—Design and Implementation

The GPP was announced in Israel on 16 November 2020, a month before the vaccination rollout began, and went into effect on 21 February 2021 [12]. At that point, about 65% of the population for whom the vaccine had been approved (those aged 16+) had received a first dose of the vaccine and 45% had received a second dose [13] (see Figure 1). By 1 May 2021, of 9.3 million Israeli residents, five million fully vaccinated and another 830,000 recovered were issued a GPP, representing about 63% of the entire population, and 91% of the population aged 16+.

At that point (May 2021), the Green Pass was an official document only for internal use in Israel and applied solely to non-essential services. The Ministry of Health (MoH) issued it to any individual who had a vaccine or disease-recovery certificate seven days after the second dose to any vaccinated resident, after a PCR test confirming recovery from a COVID-19 infection, or a positive serology test result. All GPP documents are valid until the end of 2021. Children, as well as adults with vaccine contraindications, could enter GPP-required places with a negative diagnostic PCR test that was performed up to 72 h prior to entrance. 

The GPP regulations developed and changed according to the risks (see Table 1). On 1 May 2021, the GPP allowed entrance to indoor public places and businesses mandated by the GPP requirements: university lecture halls, gyms, swimming pools, cultural and sporting events (and venues), restaurants, hotels, and clubs. The policymakers who developed the GPP considered the risk of infection and differentiated between seated events with no food served and events in which there was changing or mixing of crowds, or where food was served (such as weddings, clubs, standing concerts, public sporting events). Stricter occupancy limits applied to the latter. The use of face masks and physical distancing was still mandatory, even with the Green Pass. The GPP was coercive, as only those who had it could enter the designated places. Outdoor public spaces, malls, libraries, and museums did not require the GPP but operated according to limited occupancy and density. Houses of worship could voluntarily choose to adhere to the GPP since they were considered an essential service.

The GPP was implemented through a certificate that was easily accessible to residents who could not speak Hebrew, and could also be issued in Arabic, Russian or English. It was also accessible to technology-disadvantaged residents and could be issued by phone, email, post, or fax, as well as through the MoH’s website (https://corona.health.gov.il/en/green-pass/) (accessed on 18 October 2021). Nonetheless, the certificate could and was mainly used as a mobile phone app developed by the MoH and based on its vaccination database. The app showed the certificate as an “animation” (moving image). Since it was not a static form or a Quick Response (QR) code, it was difficult to forge.

## 4. Discussion

### 4.1. The Green Pass Policy Can Be an Effective Public Health Tool to Relax COVID-19 Restrictions in a Situation of Great Uncertainty

Evidence shows that the benefits of COVID-19 vaccines outweigh the risks [14]. The early, broad vaccination coverage in Israel and the UK provided clear evidence that the vaccine reduces severe illness and death [15,16,17,18], but the extent to which it reduces transmission is not yet confirmed [19,20,21,22], particularly given the different characteristics of the virus variants. Vaccine hesitancy has declined in many countries, yet coverage still seems to be insufficient to allow a smooth return to “normal” life [23]. Therefore, a cautious relaxation of COVID-19 restrictions is important to avoid new surges until populations approach herd immunity. The GPP is a powerful tool toward attaining this objective through the stricter movement of unprotected people, and the great incentive it creates for the hesitant to get vaccinated. In February 2021, the tool speeded up the easing of COVID-19 restrictions. This is valuable due to the high burden (economic, psychological, social, and indirect health costs) of such restrictions. In addition, the GPP created “safe areas” when easing public health restrictions and provided a sense of increased protection for individuals. Limiting social activity with the GPP had an added value because people have tended to reduce their conscientiousness and awareness after their first dose of a vaccine; this might result in higher infection rates, as the level of protection is not complete [24]. The MoH overcame these concerns by issuing the GPP one week after the second dose. After evidence emerged regarding the level of antibodies after a person received a vaccine, the policy changed, and the GPP started being issued two weeks after a person’s second dose.

Israel started the vaccine rollout on 20 December 2020, only administering the BNT162b2 Biontech-Pfizer vaccine. The vaccination rollout was remarkably fast [25,26]. In less than one month, on 2 February 2021, the prioritisation policy was opened to all age groups (Figure 2) and morbidity sharply declined (see Figure 1 and Figure 2). Since there was no data about the level of vaccination coverage that would allow herd immunity, Israel had to set its own benchmarks for vaccination and recovered rates to start relaxing COVID-19 restrictions in a situation of great uncertainty. The GPP outlined national standards for operating business and public gatherings when nearly half of the target population was vaccinated.

The GPP is a public health tool that can be implemented and lifted according to the specific epidemiological needs of the society: as vaccination rates increase, and morbidity and mortality rates decrease, governments can ease COVID-19 restrictions, including the use of the GPP, and reintroduce it during cases of new waves due to virus mutations (such as Delta), or in specific areas with “pockets” of unvaccinated people. It may prevent or at least delay the need for stricter measures such as lockdowns because it limits social activities to protected individuals, and this is expected to contain the number of severely ill COVID-19 patients. It is important to note that there is not yet clear-cut evidence that the GPP reduced morbidity loads, but since it reduces the transmission of the virus, we assume that it supports the easing of COVID-19 restrictions along with reduced morbidity.

### 4.2. What Policymakers Should Be Aware of When Considering a “Green Pass Policy”

COVID-19 has been changing the way vaccine certificates are used; there has been great debate on their advantages and disadvantages and the ethical considerations surrounding them. Recently, GPPs have become an increasingly accepted concept, particularly because restrictions on fundamental civil rights are less justifiable from a public health rationale for those who are already protected from the pandemic [27,28,29]. The Israeli experience highlights issues not yet explored in the literature (for a summary of the issues policymakers should consider when shaping a vaccine certificate policy, see Conclusions below).

Any GPP must be implemented in a timely manner. In Israel, it was developed and put into practice in a record time of two months, in part because the four health plans operating in Israel covering the entire population have standardised, universal electronic medical records with the information about the vaccine or recovery status of all residents. These records were easily compiled into one single database created by the MoH for the GPP, as privacy and data protection laws in Israel allow the sharing of this data [25]. To ensure data privacy and protection, any data related to the vaccine or recovery was only transferred if a person requested the GPP or vaccine/recovery certificate. Data was encrypted during the entire process.

The GPP raises equity concerns due to the fragmented recovery of civil rights within a country (those unvaccinated remain restricted) and the exclusion of those who cannot get the vaccine, including children, and those who cannot prove that they have recovered [27,28,29]. Israel has attempted to mitigate these equity concerns first by implementing the GPP only after the prioritisation of the vaccine was opened to all eligible populations (illustrated in Figure 1). This meant that all residents eligible and willing (in light of FDA and MoH approval, at the time, ages 16+) to receive the vaccine had the option to do so (or not), and no resident aged 16+ would be denied a GPP due to ineligibility or a lack of supply of the vaccine. Second, Israel only implemented the system after the vaccination rate was already high (about 45% of the aged 16+ fully vaccinated and 65% with the first dose). By August 2021, most high-income countries had opened eligibility to the vaccine to all age groups (older than 12, the age for which the vaccine was subsequently approved), yet middle- and low-income countries were still facing supply limitations and retained prioritisation policies. As long as not all population groups are eligible or have access to the vaccine, a GPP may not be an equitable tool [30]. In this case, negative tests were temporary solutions for the unvaccinated, granting them a temporary GPP. It is also important to include children younger than 12, who currently cannot be vaccinated, into the GPP. However, this solution is only relevant if tests are widely available and free of charge. Those who cannot get the vaccine are registered at the MoH and can receive antigen or PCR tests free of charge and a temporary GPP. Since the objective of the GPP is to reduce transmission of the virus, and those who cannot get vaccinated are more likely to transmit it more than those who are vaccinated, this cohort is treated as unvaccinated. Nevertheless, there are very few people (about 2000) within this group, and they receive their exemption on a case-by-case basis.

A second caveat is that essential activities should be excluded from the GPP requirement. For example, in Israel, in-person higher education was allowed for those with a GPP; the contingent of those who did not have it could participate virtually in all classes. A GPP is mandatory for customers of a business, but not for workers. However, some workplaces voluntarily adopted the GPP. Policymakers should be aware that there will in practice be gaps between the GPP regulations and implementation. While some places might demand it without being legally required to do so, others will not implement it despite being legally compelled to. In practice, not all businesses checked their customers or did not comply with the capacity of people allowed. The GPP policy is difficult to enforce, and compliance depends a great deal on culture—and the willingness of service providers and customers to follow suit. Creating simple and uniform rules that make epidemiological sense increases the likelihood of compliance with a GPP.

Third, GPP regulations should be implemented gradually, flexibly, and according to current risks and morbidity rates. Some degree of caution was needed in Israel even when only those with a GPP started circulating, and public gatherings remained restricted. Yet, after one month of low COVID-19 incidence, the GPP exempted non-essential outdoor public venues. For example, while individuals without a GPP would not be allowed in restaurants, swimming pools, or sports venues indoors, they were nevertheless allowed in outdoor areas s as an alternative. On 21 July 2021, Israel reintroduced the GPP, but only for closed places with more than 100 people. The GPP restrictions were further tightened on 29 July and 8 August in light of developments stemming from the Delta variant surge (for more details of adjustments to the regulations, see Table 1).

Fourth, while the GPP allowed for relaxing the COVID-19 restrictions, it kept many in place, such as occupancy restrictions. This means that the GPP still incurred costs similar (albeit lower) to those of COVID-19 restrictions: business, restaurants, public places, and venues continued to operate at partial capacity, and still needed financial support from the government to compensate for income loss. Some, such as cinemas, were even forced to permanently close because partial activity failed to cover operating costs. GPP regulations therefore should balance epidemiological risks with the economic viability of their implementation. The Israeli GPP was designed to be feasible for a long period of time. Policymakers took a participatory approach and consulted with public representatives from all sectors that would be affected by the GPP. It organised round tables to discuss the GPP design and regulations in various industries, including retail and trade, culture, sports, education, public and private events, hospitality and tourism, as well as houses of worship. These representatives raised concerns that allowed policymakers to accommodate as many requests as possible while maintaining the necessary precautions. It is also important to understand public opinion on this type of policy. In one survey, Israelis felt frustrated with the relative lack of autonomy caused by the GPP (compared to their British counterparts, where no GPP was in place) [31].

Finally, when designing and implementing a GPP, it is important to have an effective enforcement plan. For example, this would include frequent checks of whether a GPP is requested in public places and contingent on fines or other enforceable measures. Without enforcement, a GPP has little impact.

## 5. Conclusions

Summarising, policymakers should consider the following issues when designing a vaccine certificate policy:Electronic medical records can facilitate the implementation of a GPP, but attention should be paid to data privacy and protection.To reduce equity concerns, all age groups should be eligible for the vaccine (contingent on approval for children) and have access to the vaccine. Alternatively, authorities can grant temporary certificates if testing is available.Essential activities should be excluded from the GPP requirement. Nevertheless, it is impossible to avoid gaps between GPP regulations and implementation. While some places might demand a GPP even when not legally necessary, others will not implement it despite being legally obligated to do so.GPP regulations should have standardised epidemiological criteria, a gradual implementation process, flexibility in their application, and adaptability according to risk.GPP limitations should balance epidemiological risks with the economic viability of their implementation; otherwise, businesses may not recover if partial activity does not cover operating costs.Consulting with stakeholders can reduce resistance or opposition to a GPP.Enforcement is key for the effectiveness of a GPP and should be designed in a way that does not violate civil rights.

## Figures and Tables

**Figure 1 ijerph-18-11212-f001:**
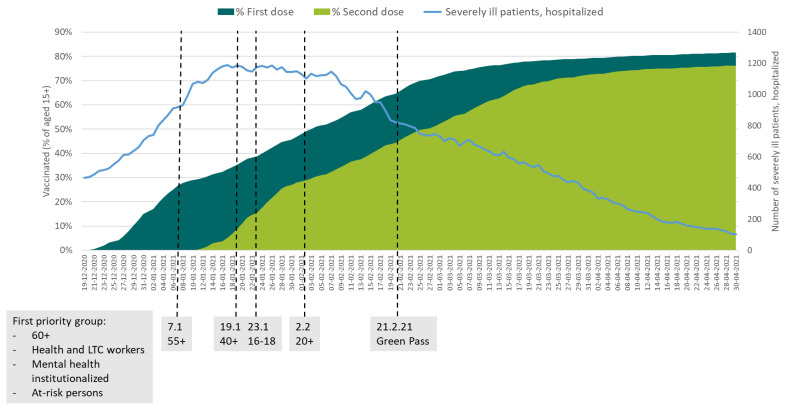
Vaccination rollout, hospitalized, prioritisation, and implementation of the Green Pass. (Sources: authors’ own compilation based on https://datadashboard.health.gov.il/COVID-19/general, accessed on 18 October 2021 and [12]; Note: vaccination rates for the population aged 15+ for whom the vaccine was approved by the manufacturer.

**Figure 2 ijerph-18-11212-f002:**
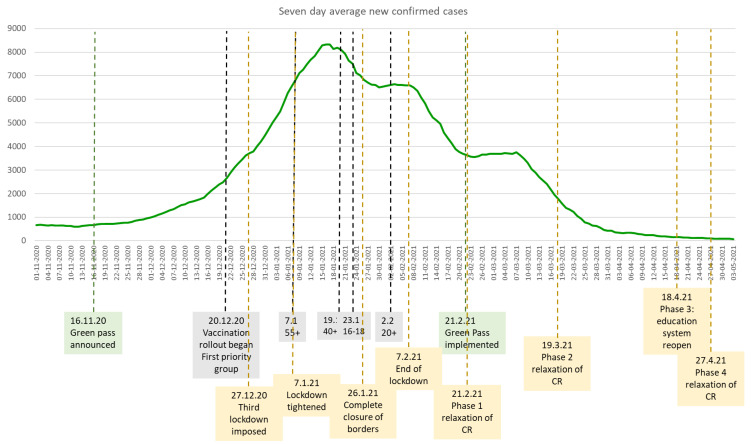
COVID-19 morbidity, vaccination rollout, public health responses, and the Green Pass. (Note: CR = COVID restrictions. Sources: Authors’ own compilation based on https://datadashboard.health.gov.il/COVID-19/general, accessed on 18 October 2021 and [12]).

**Table 1 ijerph-18-11212-t001:** Development of GPP regulations.

Phase	Date	Limitations on Gatherings	High-Risk Places	Low-Risk Places	Notes
First implementation of the GPP	21.02.21	Maximum people: 10 indoors 20 outdoors	Remained closed	75% capacity allowed, 300 indoors, 500 outdoors	Occupancy restrictions of 1 person per 7 sq. metres
Opening high-risk events and relaxing restrictions on low-risk events	07.03.21	No change	50% capacity, maximum 300 people. For restaurants capacity 75%, max. 100 people indoors, 100 outdoors	75% capacity allowed, 500 indoors, 750 outdoors	Babies up to 1 year old exempt from GPP
Further easing of restrictions	19.03.21	No change	50% capacity, maximum 300 people indoors, 500 outdoors	Indoors: 75% capacity for places with up to 5000 seats, and 30% capacity otherwise.Outdoors: 75% capacity for places with up to 10,000 places, 30% otherwise	Temporary GPP granted for negatively tested with rapid tests (privately funded). Swimming pools and events in open areas exempt from GPP
Further easing of restrictions	08.04.21	Maximum people: 20 indoors 100 outdoors	Maximum 300 people indoors, 750 outdoors	Up to 10,000 people at outdoor and 4000 indoor events	Introduction of government-funded PCR tests for children to obtain a temporary GPP
Further easing of restrictions	06.05.21	Maximum people: 50 indoors 500 outdoors	End of all occupancy restrictions	End of all occupancy restrictions	Exemptions for sporting facilities and events from the GPP
Abolition of GPP regulations and all occupancy and gathering restrictions	01.06.21				
Reintroduction of the “Green Pass Policy” for all individuals ^1^	21.07.21	No limitations	Green Pass required for indoor social events with 100+ people	No limitations	Heavy penalties imposed on owners of businesses that do not require the Green Pass
Tighter regulations for the “Green Pass Policy” ^2^	29.07.21	No limitations	Expansion of the GPP requirements to places with 100+ people such as restaurants, gyms, sporting facilities, hotels	No limitations	Children aged 12+ are exempt from the Green Pass (except social events with 100+ people). Negative tests can be used as “temporary Green Pass”: PCR for 72 h, rapid test for 24 h.
Expansion of the requirement of the Green Pass to all public places, except essential places, including children of all ages ^3^	08.08.21	No limitations	Green Pass required for all public places regardless of the number of people.	Synagogues with more than 50 people must comply with the GPP	Ending of government-funded tests; tests to be covered by the recipient, except for children up to 12 years old

^1^ https://www.gov.il/he/departments/news/16072021-01, accessed on 18 October 2021. ^2^ https://www.gov.il/he/departments/news/29072021-02, accessed on 18 October 2021. ^3^ https://www.gov.il/he/departments/news/05082021-04, accessed on 18 October 2021.

## Data Availability

Publicly available datasets were analysed in this study. This data can be found here: https://datadashboard.health.gov.il/COVID-19/general (accessed on 18 October 2021).

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
