# Peer review of "The Israeli Experience with the “Green Pass” Policy Highlights Issues to Be Considered by Policymakers in Other Countries"

_ijerph, 2021, doi:10.3390/ijerph182111212_

Round 1
Reviewer 1 Report
The authors of the paper describe the design and implementation of the Green Pass Policy (GPP) in Israel. The authors also highlight some gaps and caveats that should be considered by policymakers when planning to implement a policy similar to the GPP.
Major comment:
Lines 61-64: Figure 1 is not an accurate representation of the statement made in lines 61-64. The statement in lines 61-64 states the percentages of the population who received the first and the second dose and for whom the vaccine was approved. Whereas, the Y-axis of Figure 1 shows the percentage vaccinated from the entire population, and so the percentages of the population that received 1st and 2nd dose as shown in Figure 1 is lower than what is stated in lines 61-64. Therefore, the authors should either change the figure to have the Y-axis represent percent of population for whom vaccine was approved or revise lines 61-64 to represent the Figure 1 accurately.
Minor comment: Check manuscript for typos and minor grammatical issues. For example:
Line 49: Did the authors mean “rationale”?
Line 157: should be “in a record time”
Line 190: Did the authors mean “complied with” instead of “did not comply with”? Please rephrase to clarify what you mean here. Seems like a run-on sentence
Line 192: Typo in “costumers”. Change to “customers”
Line 200: Replace “in outdoors” with “outdoors”
Author Response
Dear editor, dear reviewers,
We thank you for the thorough review of our manuscript, and for the comments that have helped us improving it. We have revised the manuscript based on the comments, and have addressed the comments with point-by-point responses below.
Reviewer 1
Major comment:
Lines 61-64: Figure 1 is not an accurate representation of the statement made in lines 61-64. The statement in lines 61-64 states the percentages of the population who received the first and the second dose and for whom the vaccine was approved. Whereas, the Y-axis of Figure 1 shows the percentage vaccinated from the entire population, and so the percentages of the population that received 1st and 2nd dose as shown in Figure 1 is lower than what is stated in lines 61-64. Therefore, the authors should either change the figure to have the Y-axis represent percent of population for whom vaccine was approved or revise lines 61-64 to represent the Figure 1 accurately.
Response:
We have revised the figure, and it now presents the rates of vaccinated from the population aged 15+, which corresponds to the population group for which the vaccine was approved at that point in time. (In Israel the vaccine was approved for children aged 12+ on June 6th, 2021).
Minor comment: Check manuscript for typos and minor grammatical issues. For example:
Line 49: Did the authors mean “rationale”?
Line 157: should be “in a record time”
Line 190: Did the authors mean “complied with” instead of “did not comply with”? Please rephrase to clarify what you mean here. Seems like a run-on sentence
Line 192: Typo in “costumers”. Change to “customers”
Line 200: Replace “in outdoors” with “outdoors”
Response: We thank the reviewer for the proposed corrections. We have corrected the typos mentioned above, and the manuscript underwent professional English language editing.
Reviewer 2 Report
The main question addressed by the research is the importance of the "green pass" in Israel and the possible application in other countries. It is relevant and interesting, the text is clear and easy to read but needs minor spelling changes.
The GPP is very useful, I agree.
But, with current vaccines, vaccinated and protected people are vectors for the transmission of the virus. So what happens to people who cannot get the vaccine? These people must also have rights, right? Is a negative PCR in 72 hours enough? Why?
Overall, a bit of discussion is lacking.
Author Response
Dear editor, dear reviewers,
We thank you for the thorough review of our manuscript, and for the comments that have helped us improving it. We have revised the manuscript based on the comments, and have addressed the comments with point-by-point responses below.
The main question addressed by the research is the importance of the "green pass" in Israel and the possible application in other countries. It is relevant and interesting; the text is clear and easy to read but needs minor spelling changes.
Response: thank you for the positive comment. The manuscript underwent professional English language editing, and corrections have been made based on reviewer’s 1 comments .
The GPP is very useful, I agree. But, with current vaccines, vaccinated and protected people are vectors for the transmission of the virus. So, what happens to people who cannot get the vaccine? These people must also have rights, right? Is a negative PCR in 72 hours enough? Why?
Response: Thank you for this important comment. We have added a discussion about the policy towards those who cannot be vaccinated, in page 7, lines 204-209:
“Those who cannot get the vaccine are registered at the MoH, and can receive antigen or PCR tests free of charge and a temporary GPP. Since the objective of the GPP is to reduce transmission of the virus, and those who cannot get vaccinated are more likely to transmit it more than those who are vaccinated, this cohort is treated as unvaccinated. Nevertheless, there are very few people (about 2,000) within this group, and they receive their exemption on a case-by-case basis”.
Overall, a bit of discussion is lacking.
Response: Based on the reviewer’s comment, we have restructured the paper and highlighted the discussion sections. We have also added two points to the discussion. The first relates to the lack of evidence regarding the effectiveness of the GPP (lines 171-175):
“It is important to note that there is not yet clear-cut evidence that the GPP reduced morbidity loads, but since it reduces the transmission of the virus, we assume that it supports the easing of COVID-19 restrictions along with reduced morbidity.”
The second relates to the enforcement of the GPP (lines 255-258):
“Finally, when designing and implementing a GPP, it is important to have an effective enforcement plan. For example, this would include frequent checks of whether a GPP is requested in public places, and contingent on fines or other punitive or enforceable measures. Without enforcement, a GPP has little impact”.

Reviewer 3 Report
This article discusses about the use of the “Green Pass Policy” (GPP) in Israel to incentivize vaccination and to allow a safe relaxation of COVID-19 restrictions.
Although it could be of interest to the readers, I believe that it should be improved before publication.
The major concern is related to the structure of the publication. Reading the author's guidelines (https://www.mdpi.com/journal/ijerph/instructions), it is not reported the possibility to submit a brief report. For this reason, I suggest re-organizing the manuscript with a classic structure (introduction, material, and methods, results, discussion, and conclusion). In the abstract section, I encourage the same division.
For example, in this regard, the introduction should be important to introduce the thematic, inserting the study's aims. Particularly, I suggest improving the description of the aims of the study.
Minor points
- Please check the references style: it should be in accordance with the guidelines for authors. Please, check it.
Author Response
Dear editor, dear reviewers,
We thank you for the thorough review of our manuscript, and for the comments that have helped us improving it. We have revised the manuscript based on the comments, and have addressed the comments with point-by-point responses below.
Reviewer 3
This article discusses about the use of the “Green Pass Policy” (GPP) in Israel to incentivize vaccination and to allow a safe relaxation of COVID-19 restrictions.
Although it could be of interest to the readers, I believe that it should be improved before publication.
The major concern is related to the structure of the publication. Reading the author's guidelines (https://www.mdpi.com/journal/ijerph/instructions), it is not reported the possibility to submit a brief report. For this reason, I suggest re-organizing the manuscript with a classic structure (introduction, material, and methods, results, discussion, and conclusion). In the abstract section, I encourage the same division.
For example, in this regard, the introduction should be important to introduce the thematic, inserting the study's aims. Particularly, I suggest improving the description of the aims of the study.
Response: Thank you for pointing out this important comment. We have reorganized the abstract and the manuscript according to the journal’s guidelines, as per the recommendation of the reviewer. Both include now all the sections and comply with the format.
Minor points
- Please check the references style: it should be in accordance with the guidelines for authors. Please, check it.
Response: We have revised the references style to fit the journal’s guidelines.

Round 2
Reviewer 3 Report
Following the reviewers' suggestions, the authors have improved the manuscript. Nevertheless, I have several minor suggestions:
- lines 286-289 "Finally, when designing and implementing a GPP, it is important to have an effective enforcement plan. For example, this would include frequent checks of whether a GPP is requested in public places and contingent on fines or other punitive or enforceable measures. Without enforcement, the GPP has little impact."
This is a tricky phrase. In a democratic state, there are several important concerns about "punitive or enforceable measures."
I suggest rewording this phrase or erasing it.
Finally, several considerations about these concerns should be made in the conclusions.
Author Response
ijerph-1395077
Comments from reviewer #3, second round
On behalf of the authors, I thank the reviewer for the thorough read of the paper, and for the additional comments. Please see below the final comments followed by responses to each.
We hope that now the paper is suitable for publication.
Kind regards.
Following the reviewers' suggestions, the authors have improved the manuscript. Nevertheless, I have several minor suggestions:
- lines 286-289 "Finally, when designing and implementing a GPP, it is important to have an effective enforcement plan. For example, this would include frequent checks of whether a GPP is requested in public places and contingent on fines or other punitive or enforceable measures. Without enforcement, the GPP has little impact."
This is a tricky phrase. In a democratic state, there are several important concerns about "punitive or enforceable measures."
I suggest rewording this phrase or erasing it.
Response: thank you for this comment. Even in a democratic country, enforcement of regulation is possible, to the extent it is in the proper proportion. We are aware that the sentence might be too assertive, therefore we dropped the word “punitive”, leaving only “contingent on fines or other enforceable measures”. This way, each country can find its own enforceable measures.
Finally, several considerations about these concerns should be made in the conclusions.
Response: We believe that if the GPP is designed according to the recommendations mentioned in the discussion and conclusions, it does not violate civil rights and can be implemented in democratic countries. In fact, there are many democratic countries that have been implementing such a policy, without undermining basic rights. Nevertheless, we agree that these concerns should be highlighted in the conclusions, and we have added some considerations about concerns of violation of civil rights with the enforcement of the GPP:
“Enforcement is key for the effectiveness of a GPP, and should be designed in a way that does not violate civil rights.”
